# Hydroxytyrosol Supplementation Modifies Plasma Levels of Tissue Inhibitor of Metallopeptidase 1 in Women with Breast Cancer

**DOI:** 10.3390/antiox8090393

**Published:** 2019-09-11

**Authors:** Cesar Ramirez-Tortosa, Ana Sanchez, Cristina Perez-Ramirez, Jose Luis Quiles, María Robles-Almazan, Mario Pulido-Moran, Pedro Sanchez-Rovira, MCarmen Ramirez-Tortosa

**Affiliations:** 1UGC de Anatomía Patológica Hospital San Cecilio de Granada, Avda. Conocimiento s/n, 18071 Granada, Spain; cesarl.ramirez.sspa@juntadeandalucia.es; 2Department of Biochemistry and Molecular Biology II, Faculty of Pharmacy, University of Granada, 18071 Granada, Spain; anasf@correo.ugr.es (A.S.); cperezramirez87@gmail.com (C.P.-R.); mpulido87@gmail.com (M.P.-M.); 3Institute of Nutrition and Food Technology, Biomedical Center Research, Avda. Conocimiento s/n, 18071 Granada, Spain; jlquiles@ugr.es; 4Department of Physiology, Faculty of Pharmacy, University of Granada, 18071 Granada, Spain; 5Department of Medical Oncolgy, Jaen Hospital, 23007 Jaen, Spain; maria.robles.exts@juntadeandalucia.es (M.R.-A.); oncopsr@yahoo.es (P.S.-R.)

**Keywords:** hydroxytyrosol, antioxidant, extra virgin olive oil, TIMP-1, MMP-9

## Abstract

The etiology of breast cancer can be very different. Most antineoplastic drugs are not selective against tumor cells and also affect normal cells, leading to a wide variety of adverse reactions such as the production of free radicals by altering the redox state of the organisms. Therefore, the objective of this study was to elucidate if hydroxytyrosol (HT) (an antioxidant present in extra virgin olive oil) has a chemomodulatory effect when combined with the chemotherapeutic drugs epirubicin and cyclophosphamide followed by taxanes in breast cancer patients. Changes in plasma levels of matrix metalloproteinase 9 (MMP-9) and tissue inhibitor of metalloproteinases 1 (TIMP-1) throughout the chemotherapy treatment were studied. Both molecules are involved in cell proliferation, apoptosis, neoangiogenesis, and metastasis in breast cancer patients. Women with breast cancer were divided into two groups: a group of patients receiving a dietary supplement of HT and a control group of patients receiving placebo. The results showed that the plasma levels of TIMP-1 in the group of patients receiving HT were significantly lower than those levels found in the control group after the epirubicin-cyclophosphamide chemotherapy.

## 1. Introduction

Most antineoplastic drugs are not selective against tumor cells; they also affect normal cells, leading to a wide variety of adverse events in some tissues of the body. These adverse events are derived from the mechanism of action of these drugs, including the production of free radicals, which affect the redox state of the organism [1,2].

Oxidative stress is the consequence of an imbalance in the redox state. The increase in reactive oxygen species (ROS) and free radicals generated in multiple metabolic pathways contributes to this imbalance due to an increase in oxidation, which can lead to tissue damage [3].

These ROS represent an important factor in carcinogenesis and can play a role in the three stages of cancer: initiation, promotion, and tumor progression. Free radicals cause oxidative damage in the DNA, contributing to the mutagenesis, which is essential for the process of tumor initiation. This damage caused by free radicals can be minimized by enzymes such as catalase, superoxide dismutase or glutathione peroxidase; or by other non-enzymatic antioxidant mechanisms (vitamins A, C and E, selenium and reduced glutathione (GSH) [4] and maximized by cytochrome P450, xanthine oxidase, and NADPH oxidases [5].

In the stage of cancer promotion, ROS can interact with surface or intracellular receptors (tyrosine kinases receptor (RTKs), thus modulating signaling pathways (MAPK-and PI3 Kinase dependent) and physiological mechanisms related to proliferation, apoptosis, angiogenesis, and others [6].

The malignant transformation of tumor cells is generally characterized by an increase in motility, invasiveness, genetic instability and angiogenesis. These characteristics are acquired during the stage of tumor progression [5]. There is evidence that ROS can promote the stabilization of alpha subunit of hypoxia-inducible factor (HIF-1α), crucial in the process of neovascularization and angiogenesis [7]. Other studies reveal that NADPH oxidase 1 (NOX-1), which catalyzes the production of ROS, can promote angiogenesis through the regulation of receptors for vascular endothelial growth factor (VEGF) and the activity of matrix metalloproteinases (MMPs) [8].

Hydroxytyrosol (HT) is a polyphenol with a phenethyl alcohol structure. The chemical name of HT is 3,4-dihydroxyphenylethanol. This compound is in minor amount in extra virgin olive oil, in particular, in the water-soluble fraction. Hydroxytyrosol is generated from the hydrolysis of oleuropein, this process occurs during the maturation of olives, the storage of the oil and the preparation of table olives. This compound is also present in olive leaves and in different types of wine at various concentrations (higher in red wine) [9,10,11].

Numerous studies have been conducted with isolated HT and with various olive oils rich in HT. The antioxidant, anti-inflammatory, and antiatherogenic effects of HT have been demonstrated, as well as its role in the prevention and modulation of various diseases, including cancer [12]. Many of the anti-tumor properties of HT may be due to other activities, such as its ability to modulate the antioxidant system and to eliminate ROS [10,13,14]. Several studies conducted in vitro and in vivo have demonstrated the antitumor activity of HT. Granados-Principal et al. [15] showed that HT has an antitumor effect in Sprague-Dawley rats with experimental breast cancer by inhibiting cancer cell growth and proliferation. However, there are no clinical trials published to date demonstrating the antitumor effect of HT in humans with breast cancer.

MMPs are zinc-dependent proteolytic enzymes, involved in tumor invasion and metastasis, because MMPs play a fundamental role in the degradation and remodeling of the extracellular matrix. In addition, other activities have been attributed to MMPs, such as participating in the regulation of cell proliferation and the release of growth factors and contributing to the angiogenesis characteristic of tumor cells [16,17].

The expression of MMP-9 has been shown to be induced by transcription factors, such as Fos and Jun, both produced through the Ras/Raf/ERK pathway, which is known to be an activating pathway for cell proliferation [18]. Several authors have associated higher plasma levels of MMP-9 with a worse prognosis. Thus, Scarpa et al. [19] and Lawicki et al. [20] found a direct association between high plasma levels of MMP-9 and a lower tumor differentiation, associating with a worse prognosis.

Furthermore, tissue inhibitor of metalloproteinases (TIMPs) are a family of proteins capable of inhibiting the activity of MMPs through their binding to the catalytic site through non-covalent bonds. Thus, TIMPs play a key role in maintaining the balance between the deposition of the extracellular matrix and its degradation, this role is played through the binding to the MMPs and, consequently, the regulation of the activity of MMPs. In addition, other functions have also been attributed to TIMPs, such as participating in cell proliferation, apoptosis, angiogenesis, tumor invasion, and metastasis [21,22].

Some studies have showed a relationship between higher TIMP-1 levels and a worse prognosis in patients with breast cancer. Lawicki et al. [23] found higher plasma levels in advanced cancer stages, thus being able to establish a relationship with a worse prognosis. The same conclusions were found by Lawicki et al. [20] and by Schrohl et al. [24], who showed that the plasma of women with mammary tumor metastasis presented higher levels of TIMP-1 than those women with primary tumors, showing both studies a relationship between plasma levels of TIMP-1 and more advanced stages of cancer and, therefore, a more unfavorable prognosis [22].

Based on the above, the objective of the present study was to elucidate whether HT improves the antitumor response of women with breast cancer undergoing neoadjuvant chemotherapy (treated with epirubicin and cyclophosphamide followed by taxanes). We also aim to elucidate if this effect could be produced by modulating the plasma levels of MMP-9 and TIMP-1, leading to a decrease in cell proliferation and, therefore, improving the prognosis of the patients.

## 2. Materials and Methods

Two experimental groups were established to investigate the effect of dietary supplementation with HT in breast cancer patients undergoing neoadjuvant chemotherapy: patients in Group A (n = 20) were supplemented with HT at a single dose of 15 mg/day administered as hard capsules; and patients in Group B (control group n = 20) received a placebo hard capsule. The placebo capsules had the same pharmaceutical form and color as the usual format of the supplement. The manufacturing entity was the only one that kept information about the packaging corresponding to the supplement and the placebo. This information was only revealed once the study was completed. The capsules used in the study were supplied by Probelte Pharma S.A. (Murcia, Spain) following all the quality, stability, conservation, and labeling controls for human dietary supplements. Placebo capsules consisted of 96% Eliano MD2 (maltodextrin), 2% tricalcium phosphate and 2% magnesium stearate. Hydroxytyrosol capsules consisted of 63.5% Eliano MD2 (maltodextrin), 2% tricalcium phosphate, 2% magnesium stearate and 34.5% Mediteanox. Mediteanox™ is a natural extract obtained of leaves and olives from olive tree (*Olea europaea*) of Jaen grown in Spain. This extract is obtained by a patented technology, using extraction procedures based on ultrapure water. The main component of Mediteanox™ is the potent natural antioxidant HT (15 mg/capsule). Toxicology Tests support the safety of Mediteanox™.

The design of this study was a triple-blind randomized trial with parallel groups (supplemented with HT or placebo). The study population consisted of those patients selected by the Breast Committee for neoadjuvant therapy according to the criteria included in the Clinical Practice Guideline prepared by the Subcommittee at Jaen Hospital. All the patients signed an informed consent for their inclusion in the study prior to their participation. The study was conducted in accordance with the Declaration of Helsinki, and the protocol was approved by the Ethics committee of Jaen Hospital, Spain (code: PI-0695-2012).

The following inclusion criteria were considered to participate in the clinical trial: luminal (A or B) tumor phenotype, receiving neoadjuvant therapy based on anthracyclines and taxanes and signed the Informed Consent. The exclusion criteria were: metastatic disease at diagnosis, Her2-positive or triple-negative phenotypes, history of mental illness, detection in the initial interview of a psychological profile that might lead to a low adherence to the supplementation regiment, women who only received hormonal therapy as neoadjuvant treatment and women with known allergies to foods or compounds derived from the olive tree.

The overall sample size was estimated based on the data obtained in the study published by Vera-Ramírez et al. [25]. Matched blood samples were collected from each patient in T1, T2 and T3 cycles of chemotherapy. Approximately 5 mL of blood was taken from each patient, by venous puncture, drawn into an ethylene diamine tetra-acetic acid (EDTA)-containing tube (Vacutainer® EDTA Tubes; BD, Franklin Lakes, NJ, USA), and centrifuged at 1000× *g* for 15 min. The plasma was kept in a separate tube and frozen at −80 °C. The sample was recruited consecutively to its identification and sequentially. Patients were randomly assigned to one of the groups. The final number of patients included in the study was 40 (n = 20 per experimental group). The timeline of the study is shown in Figure 1. The analytical determinations were measured at three time points—T1, T2, and T3—as described in Figure 1.

The habitual diet of the patients was daily checked with 24 h dietary recalls using food records of measured and weighed food intake and all recipes of homemade dishes for one week. In particular, three recall days were registered at the day of recruitment by a dietician at T1, T2, and T3 time points. Another four days (including one weekend day) were registered by the patient, starting on the first day after recruitment, with further supervision by the dietician. The content of macronutrients and selected micronutrients in the diet was calculated using the computer program ALIMENTACION Y SALUD 0698.046 (BitASDE General Medica Farmaceutica, Valencia, Spain) (data not shown).

### 2.1. Plasma Metalloproteinase-9 (MMP-9) Assay

Plasma samples were stored at −80 °C, so before making the determinations, they were thawed gradually at 4–10 °C approximately in the refrigerator. The dilutions of the samples have always been performed in cold to maintain their integrity and to ensure reliable results.

Plasma levels of MMP-9 were measured with the kit “Enzyme-linked Immunosorbent Assay Kit for Matrix Metalloproteinase 9” from the commercial company Cloud-Clone Corp. (Cloud-Clone Corporation, Houston, TX, USA).

To perform the plasma determination of MMP-9, plasma samples were first diluted at a 1:100 dilution, using 0.01 mol/L PBS prepared extemporaneously as solvent. Then, the standards were prepared according to the kit protocol. Subsequently, once samples and diluted standards were prepared, 100 µL of each sample, the blank and the standards were added into the corresponding wells and incubated for one hour at 37 °C, after which the liquid was removed from the wells and 100 µL of Detection Reagent A (containing antibody specific against MMP-9) were added to each well, and plates were incubated again at 37 °C for one hour. Then, the plate was washed three times with the washing buffer included in the kit. Next, Detection Reagent B (containing the conjugated secondary antibody) was added and the plate was incubated at 37 °C for 30 min. After this process, the plate was washed five times with the washing buffer and 90 µL of substrate was added into each well. The plate was placed in an incubator at 37 °C for 15 min isolated from the light, after which an intense blue coloration occurred, which turned yellow after the addition of 50 µL of stop solution. Finally, the absorbance of the plate was measured in a spectrophotometer at 450 nm.

Plasma levels of MMP-9 present in the samples was obtained by entering the optical density (OD) results obtained into the online desktop tool MyAssays (www.myassays.com).

With this application, a standard curve of 4 parameters was drawn and the OD values measured in plasma samples were extrapolated, thus obtaining the levels of MMP-9 expressed in ng/mL.

### 2.2. Plasma Tissue Inhibitor of Metalloproteinases-1 (TIMP-1) Assay

Plasma levels of TIMP-1 were measured with the “Enzyme-linked Immunosorbent Assay Kit for Tissue Inhibitors of Metalloproteinase 1” from the commercial firm Cloud-Clone Corp. (Cloud-Clone Corporation, Houston, TX, USA).

Once the diluted samples and standards were prepared, 100 µL of each sample, the blank and the standards were added into the corresponding wells and incubated for one hour at 37 °C, after which the liquid was removed from the wells and 100 µL of Detection Reagent A (containing specific antibody against TIMP-1) were added to each well, and plates were incubated again at 37 °C for one hour. Then, the plate was washed three times with the corresponding washing buffer. Subsequently, the Detection Reagent B containing the conjugated secondary antibody was added and the plate was incubated at 37 °C for 30 min, then the plate was washed five times with washing buffer. The next step was the addition of 90 µL of substrate. The plate was maintained at 37 °C for 15 min in the dark, thus generating an intense blue color that turned yellow after the addition of 50 µL of stop solution. Finally, the absorbance of the plate was measured in a spectrophotometer at 450 nm.

The final concentration of TIMP-1 in the samples was determined by entering the OD results obtained into the online desktop tool MyAssays (www.myassays.com). With this application, a standard curve of four parameters was drawn and the OD values corresponding to plasma samples were extrapolated, thus obtaining the levels of TIMP-1 expressed in ng/mL.

### 2.3. Statistical Analysis

The results are expressed as mean and standard error of the mean. Before performing the statistical analysis, the normality and homogeneity of the variances were verified using the Kolmogorov-Smirnoff and Levene tests. When the variance followed a normal distribution and it was homogeneous, the Bonferroni test was used. Whereas, when the variances were not homogeneous, the nonparametric Kruskall-Wallis, U-Mann-Whitney, and Tamhane tests were used. An analysis of the variance (ANOVA) was performed to find out if there were differences among time points (T1, T2 and T3) for each experimental group (A and B). The significance was established at *p* < 0.05. The Student’s *t* test was performed to find differences between Group A and Group B for the same time point. All the statistical analyses were performed using the SPSS software version 22.0 (IBM Corp., Armonk, NY, USA). The results are shown in bar graphs. Bars not sharing superscript letters are statistically different (*p* < 0.05).

## 3. Results

### 3.1. Characteristics of the Population and Homogeneity of the Experimental Groups at the Beginning of the Study

Table 1 shows the most important clinical and anatomopathological variables of the group treated with HT and the control group of patients with breast cancer. Table 1 indicates that the groups were homogeneous at the beginning of the study because no significant differences (*p* > 0.05) were found between both experimental groups

All patients took every day the hard capsule containing HT or the placebo capsule during the neoadjuvant chemotherapy until the pre-surgery day (T1, T2, and T3) (Figure 1). No adverse reactions were caused by these capsules, and they were well accepted and tolerated by all patients. The daily food intake of the patients decreased due to the chemotherapy treatment. The intake of extra virgin olive oil was recorded in all patients throughout the study (approximately 10–15 mL/day in both groups).

### 3.2. Plasma Levels of MMP-9 in Breast Cancer Patients

Figure 2 shows plasma MMP-9 levels in Group A (HT) and in Group B (Placebo) of women with breast cancer undergoing neoadjuvant chemotherapy at the beginning of the study (T1), after epirubicin-cyclophosphamide treatment (T2) and, finally, after taxane treatment (T3). No significant differences were found between Group A and Group B at each time point (TI, T2, and T3).

However, there is a very pronounced decrease in the level of MMP-9 throughout the chemotherapy (T1-T2) (from 172.68 ± 16.5 to 109.82 ± 12.5 ng/mL for group A and from 244.91 ± 41.9 to 113.16 ± 18.6 ng/mL for group B). The Bonferroni test showed that the level of MMP-9 in T1 is significantly higher than the level found in T2 and T3 (93.11 ± 11.6 ng/mL for group A and 85.22 ± 10.2 ng/mL for group B) (*p* < 0.05) in both groups.

Moreover, a decrease in MMP-9 plasma level was found after the chemotherapy with taxane (T3), although this change was not statistically significant, which indicates that the combination of epirubicin plus cyclophosphamide is responsible for the pronounced decrease in plasma levels of MMP-9 throughout the chemotherapy course. This is of great interest, since the fact of achieving such a marked decrease in plasma levels of MMP-9 may be related to a better prognosis for patients.

### 3.3. Plasma Levels of TIMP-1 in Breast Cancer Patients

Figure 3 shows the average plasma levels of TIMP-1. The results have been obtained by differentiating the samples belonging to Group A and Group B during blood collection at different time points.

The results show that Group A presents significant differences in the plasma levels of TIMP-1 between T1 and T2 (from 194.94 ± 12.95 ng/mL to 152.93 ± 12.99 ng/mL). There is also a significant difference in TIMP-1 levels between T1 and T3 (from 194.94 ± 12.95 ng/mL to 152.98 ± 11.89 ng/mL). However, no significant difference in plasma levels of TIMP-1 was found in Group B between T1 and T2 (from 217.21 ± 18.93 ng/mL to 177.31 ± 20.30 ng/mL). Therefore, Group B only showed statistical significance for plasma levels of TIMP-1 between the time points T1 and T3 (217.21 ± 18.93 ng/mL vs. 164.77 ± 12.52 ng/mL). These values indicate that, the group treated with HT, at the time point at which patients were treated with epirubicin and cyclophosphamide, show a significant decrease in plasma levels of TIMP-1 compared to those patients who took the placebo. These results show that HT is exerting some effect on the plasma levels of TIMP-1.

## 4. Discussion

MMP-9 and TIMP-1 have been linked to the prognosis of breast cancer [21,22]. Therefore, we suggest that HT, due to its antioxidant, anti-proliferative, and anti-tumor activity, could modulate plasma levels of these biomarkers and improve the prognosis of patients. Thereby, this compound could be included as a supplement during breast cancer chemotherapy.

In this study, no significant differences in plasma levels of MMP-9 were found between the group treated with HT and the group that received the placebo. In contrast, other studies have shown that HT at a dose of 1–10 μmol/L decreased the expression of MMP-9 in cultures of bovine aortic endothelial cells [26] and monocytes, attributing it an antiangiogenic activity [27]. This inconsistency in the results could be explained by the fact that the cited study was performed on a culture of cells from vascular tissue but not on tumor cells.

Furthermore, plasma levels of both MMP-9 and TIMP-1 have decreased significantly from T1 to the end of treatment with taxane (T3), which indicates that chemotherapy acts as a modulator of the plasma levels of both proteins. However, it is noteworthy that for both proteins (MMP-9 and TIMP-1), plasma levels only dropped significantly at the end of the epirubicin-cyclophosphamide treatment (T2) but there were not significant differences between T2 and after treatment with taxane (T3).

Regarding MMP-9, we found that in a study in which doxorubicin (anthracycline), cyclophosphamide and 5-fluorouracil were administered, the authors did not find a decrease in the plasma levels of MMP-9 throughout the treatment [28]. In an assay performed on glioma cell lines, it was found that epirubicin decreased the secretion of MMP-9 by about 50%, which suggests a potential antiproliferative and antimigratory role of epirubicin in this type of cells [29]. This inhibitory role of epirubicin on MMP-9 had already been found in a study conducted by Karakiulakis et al. [30], where the activity of type IV collagenases was measured, being diminished by anthracyclines. All these studies are consistent with our results.

Moreover, the group receiving a HT supplement showed a significant decrease in TIMP-1 levels from T1 to T2, corresponding to the time interval in which patients have received different cycles of combined chemotherapy of epirubicin and cyclophosphamide. In contrast, no statistically significant differences in the plasma levels of TIMP-1 were found between T1 and T2 in the control group. To explain the observed results of HT supplement on plasma levels of TIMP-1, the authors suggest the following mechanism.

NFkB is a transcription factor activated in response to bacterial or viral stimuli, growth factor or inflammatory molecules. Activation of NFkB is a common characteristic of many tumors [31]. Moreover, some studies conducted by Illesca et al. [32] and Valenzuela et al. [33] have concluded that the activation of the transcription factors Nrf2 and PPAR-delta and the down-regulation of NFkB and SREBP-1c appear as important mechanisms mediating the beneficial effect of HT in metabolic disturbances in white adipose tissue. In addition, Liu et al. [34] stated that NFkB activation significantly increased the transcription of TNF-alfa, which can elicit NFkB signaling, subsequently causing an up-regulation of TGF-B [31]. TGF-B induces TIMP-1 gene expression in fibroblasts [35]. The above-mentioned shows that HT can downregulate NFkB and it could decrease TNF-alfa and TGF-b levels as described for other antioxidant agents, such as silymarin in rat liver injury induced by CCl4 [36] causing a decrease in plasma levels of TIMP-1 in women with breast cancer.

## 5. Conclusions

A supplementation with 15 mg/day of HT combined with chemotherapy treatment based on epirubicin plus cyclophosphamide decreases plasma levels of TIMP-1 in women with luminal subtype breast cancer, showing that the selected combination is of choice for improving the prognosis of patients with this pathology. Finally, to elucidate the exact mechanism by which HT leads to a decrease in TIMP-1 level, future research studies should be performed on clinical trials.

## Figures and Tables

**Figure 1 antioxidants-08-00393-f001:**
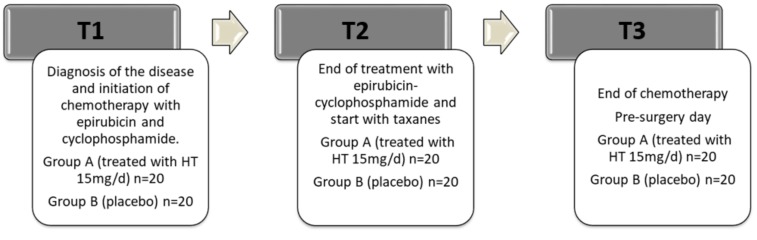
Timeline of the clinical study. **T1**: study start, total time of the period 63 days, three cycles of chemotherapy with epirubicin and cyclophosphamide, 21 days each cycle. **T2**: star of treatment with taxanes, total time of the period 63 days, three cycles of chemotherapy, 21 days each cycle. **T3**: end of chemotherapy treatment and pre-surgery day. HT dose 15 mg/d from T1 until T3.

**Figure 2 antioxidants-08-00393-f002:**
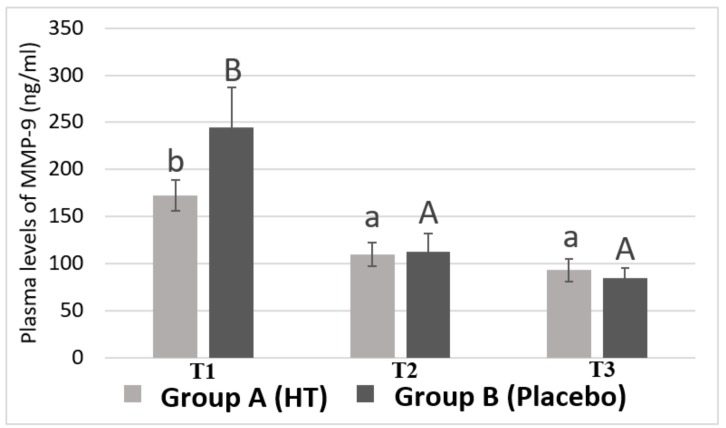
Plasma levels of MMP-9 in women with breast cancer. Values are shown as the mean ± standard error of the mean. Bars not sharing superscript letters are statistically different at *p* < 0.05 for each experimental group (lowercase for Group A treated with hydroxytyrosol and uppercase for Group B treated with placebo) at each time point (TI, T2 and T3).

**Figure 3 antioxidants-08-00393-f003:**
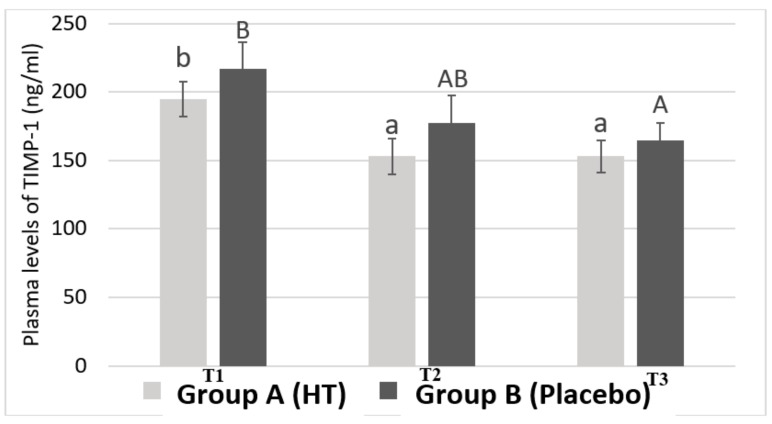
Plasma levels of the tissue inhibitor of metalloproteinases I (TIMP-1) in women with breast cancer. Values are shown as the mean ± standard error of the mean. Bars not sharing superscript letters are statistically different at *p* < 0.05 for each experimental group (lowercase for Group A treated with hydroxytyrosol and uppercase for Group B treated with placebo) at each time point (TI, T2 and T3).

**Table 1 antioxidants-08-00393-t001:** Clinical variables of the experimental groups at the beginning of the study.

Variable	Group A (Hydroxytyrosol)		Group B (Placebo)		*p* Value
		%		%	
Age (years)	51.20 ± 2.02	-	50.85 ± 1.80	-	0.80
Weight (Kg)	68.05 ± 3.13	-	68.34 ± 2.90	-	0.88
BMI	26.67 ± 1.11	-	27.57 ± 1.31	-	0.91
% Estrogen Receptor (biopsy)	92.75 ± 2.47	-	86.15 ± 5.55	-	0.53
% Progesterone Receptor (biopsy)	57.55 ± 8.02		39.8 ± 8.07		0.12
% Ki67 (biopsy)	21.79 ± 3.34	-	32.90 ± 5.11	-	0.14
Subtypes of breast cancer (biopsy)	% Luminal A	-	50	-	25
	% Luminal B	-	50	-	75

Data are expressed as mean ± SEM. BMI: Body Mass Index. *p* < 0.05 is considered statistically significant.

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
