# Peer review of "Hydroxytyrosol Supplementation Modifies Plasma Levels of Tissue Inhibitor of Metallopeptidase 1 in Women with Breast Cancer"

_antioxidants, 2019, doi:10.3390/antiox8090393_

Round 1

Reviewer 1 Report

General consideration

This manuscript is very confusing and not clear. The English is poor and difficult to follow. The rationale of the study is not clear. Why HT should modify the expression of MMP-9 and TIMP-1 in patients with breast cancer? The results are weak and not well presented. No significant physiological relevant effect was observed after treatment with HT.

Some specific concerns

Lines 35-38 needs a reference.

Line 48, 55: EROs, what does this abbreviation mean?

Line 56: VEGF, what does this abbreviation mean?

Line 63: AO, what does this abbreviation mean?

Line 69: “Their activity” who is the subject?

Table 1: The last two lines are not clear.

Figure 2: the figure is not clear. The T1, T2 and T3 are not shown. Line 232-233: What is the meaning of the sentence “The results are expressed as the interaction of time x group”. The statistical analysis is not clear: which test was used the t-Student or Bonferroni? Why were used letters instead of asterisk? Why capital letters and small letters?

Figure 3: similar to figure 2.

Author Response

We thank reviewer # 1 for constructive and valuable comments, we have modified the text according to the reviewer suggestions. All changes set out in Article have been highlighted in red ink to make them more visible. The authors acknowledge Nutraceutical Translations for English language editing of this manuscript.

Reviewer: This manuscript is very confusing and not clear. The English is poor and difficult to follow. The rationale of the study is not clear. Why HT should modify the expression of MMP-9 and TIMP-1 in patients with breast cancer? The results are weak and not well presented. No significant physiological relevant effect was observed after treatment with HT.

Authors:  According to the reviewer suggestions, authors have checked all the manuscript to improve and to clarify it. Authors have modified especially materials and methods, results, discussion and conclusion issues.

Reviewer: Lines 35-38 needs a reference

Authors:  News references have been included after lines 38.

1.            Vera-Ramirez, L.; Ramirez-Tortosa, M.C.; Perez-Lopez, P.; Granados-Principal, S.; Battino, M.; Quiles, J.L. Long-term effects of systemic cancer treatment on DNA oxidative damage: the potential for targeted therapies. Cancer Lett. 2012, 327, 134-141.

2.            Vera-Ramirez, L.; Sanchez-Rovira, P.; Ramirez-Tortosa, M.C.; Ramirez-Tortosa, C.L.; Granados-Principal, S.; Lorente, L.A.; Quiles, J.L. Oxidative stress status in metastatic breast cancer patients receiving palliative chemotherapy and its impact on survival rates. Free Radic Res. 2012, 46, 2-10.

Reviewer: Line 48, 55: EROs, what does this abbreviation mean?

Authors:  this abbreviation has been changed by ROS.

Reviewer: Line 56: VEGF, what does this abbreviation mean?

Authors:  the mean of VEGF abbreviation has been included in the manuscript. “can promote angiogenesis through the regulation of receptors for vascular endothelial growth factor (VEGF) and the activity of matrix metalloproteinases (MMPs)[7].”

Reviewer: Line 63: AO, what does this abbreviation mean?

Authors:  this abbreviation has been removed.

Reviewer: Line 69: “Their activity” who is the subject?

Authors:  According with the reviewer this sentence have been rewritten: “].  Several of studies conducted in vitro and in vivo have demonstrated the antitumor activity of HT. Granados-Principal et al. [14] showed that HT has an anti-tumor effect in Sprague-Dawley rats with experimental breast cancer by inhibiting cancer cell growth and proliferation.”

Reviewer: Table 1: The last two lines are not clear.

Authors:  Taking into account this consideration, authors have removed these lines.

Reviewer: Figure 2: the figure is not clear. The T1, T2 and T3 are not shown. Line 232-233: What is the meaning of the sentence “The results are expressed as the interaction of time x group”. The statistical analysis is not clear: which test was used the t-Student or Bonferroni? Why were used letters instead of asterisk? Why capital letters and small letters?

Authors:  Considering this kind recommendation, figure 2 has been modified and TI, T2 and T3 now are shown. “Figure 2. Plasma levels of MMP-9 in women with breast cancer. Values are shown as the mean ± standard error of the mean. Bars not sharing superscript letters are statistically different at p < 0.05 for each experimental group (lowercase for Group A treated with hydroxytyrosol and uppercase for Group B treated with placebo) at each time point (TI, T2 and T3)..”.

Reviewer: Figure 3: similar to figure 2

Authors:  Figure 3 has also been modified and TI, T2 and T3 now are shown.: “Figure 3. Plasma levels of the tissue inhibitor of metalloproteinases I (TIMP-1) in women with breast cancer. Values are shown as the mean ± standard error of the mean. Bars not sharing superscript letters are statistically different at p < 0.05 for each experimental group (lowercase for Group A treated with hydroxytyrosol and uppercase for Group B treated with placebo) at each time point (TI, T2 and T3).”

All authors have seen checked and approved the changes and suggestions exposed in this document.

Reviewer 2 Report

The manuscript submitted by Ramirez - TorTosa et al. Is an interesting clinical study that confirms the beneficial effects of hydroxytyrosol (HT) in humans. Especially in neoplastic processes. This is a remarkable aspect for this study. Because currently most of the available information is in animal or cellular models.

Regarding the manuscript, this is well written and structured. The introduction is consistent with the research proposal, the methodology used is adequate and the results are very interesting and novel. However, it is important that the authors make the following changes or improvements in the manuscript.

Major comments

1. The paragraph between lines 63 to 66 indicates important effects of HT. However, it does not have a reference.

Suggested reference:

Echeverría et al., Hydroxytyrosol and Cytoprotection: A Projection for Clinical Interventions. Int J Mol Sci. 2017; 18 (5).

2. Material and methods. The section where HT supplementation is indicated should be better written. In addition, the product is only HT ?, is an extract rich in HT ?. Explain this point.

3. The inclusion and exclusion criteria are not presented as they are written in the manuscript. They are usually presented in a paragraph and not separated by periods.

4. Figure 2 and 3. Plasma level of X ????. Replace with MMP-9 and TIMP-1.

5. Do the authors have information on the intake of foods with a high HT content (for example, Extra Virgin Olive Oil)? It would be very interesting to include this information.

6. The authors have data on oxidative stress and plasma antioxidant capacity. It would be very good to complement the obtained results.

7. HT as a cytoprotective agent has been shown to be an efficient regulator of the activity of transcription factors such as Nrf2 and NF-kB in liver and adipose tissue. What can the authors discuss in this regard?

Suggested references:

Illesca et al., Hydroxytyrosol supplementation ameliorates the metabolic disturbances in white adipose tissue from mice fed to high-fat diet through recovery of transcription factors Nrf2, SREBP-1c, PPAR-γ and NF-κB. Biomed Pharmacother. 2019; 109: 2472-2481.

Valenzuela et al., Molecular adaptations underlying the beneficial effects of hydroxytyrosol in the pathogenic alterations induced by a high-fat diet in mouse liver: PPAR-α and Nrf2 activation, and NF-κB down-regulation. Food Funct. 2017; 8 (4): 1526-1537.

8. What is or would be the possible mechanism that would be explaining the observed results?

9. What is the projection of the study ?. Discuss this point.

Minor comments:

1. Improve the title, because the use of ",". In scientific English it is very little used.

2. Improve the wording of the study objective

3. Improve the legends of tables and figures.

4. Do the authors have information on the tolerance of the product used?

Author Response

We thank reviewer # 2 for constructive and valuable comments, we have modified the text according to the reviewer suggestions. All changes set out in Article have been highlighted in red ink to make them more visible. The authors acknowledge Nutraceutical Translations for English language editing of this manuscript.

Reviewer: Comments and Suggestions for Authors

The manuscript submitted by Ramirez - Tortosa et al. Is an interesting clinical study that confirms the beneficial effects of hydroxytyrosol (HT) in humans. Especially in neoplastic processes. This is a remarkable aspect for this study. Because currently most of the available information is in animal or cellular models.

Regarding the manuscript, this is well written and structured. The introduction is consistent with the research proposal, the methodology used is adequate and the results are very interesting and novel. However, it is important that the authors make the following changes or improvements in the manuscript.

Authors:  According to the reviewer suggestions, authors have checked all the manuscript to improve and to clarify it. Authors have modified especially materials and methods, results, discussion and conclusion issues.

Major Comments

Reviewer: The paragraph between lines 63 to 66 indicates important effects of HT. However, it does not have a reference. Suggested reference:

Echeverría et al., Hydroxytyrosol and Cytoprotection: A Projection for Clinical Interventions. Int J Mol Sci. 2017; 18 (5).

Authors:  Taking into account this consideration, authors have included a new reference. 11.     Echevarría, F.; Ortiz, M.; Videla, L.A. Hydroxytyrosol and cytoprotectin: a projection for clinical interventions. Int J Mol Sci. 2017,18, 930.

Reviewer: Material and methods. The section where HT supplementation is indicated should be better written. In addition, the product is only HT?, is an extract rich in HT ?. Explain this point

Authors: According to reviewer’s suggestion, authors have included in materials and methods issue more information about the capsules used in the study. “…Two experimental groups were established to investigate the effect of dietary supplementation with HT in breast cancer patients undergoing neoadjuvant chemotherapy: patients in Group A were supplemented with HT at a single dose of 15 mg / day administered as hard capsules; and patients in Group B (control group) received a placebo hard capsule. The placebo capsules had the same pharmaceutical form and color as the usual format of the supplement. The manufacturing entity  was the only one that kept information about the packaging corresponding to the supplement and the placebo. This information was only revealed once the study was completed. The capsules used in the study were supplied by Probelte Pharma S.A. (Murcia, Spain) following all the quality, stability, conservation and labeling controls for human dietary supplements. Placebo capsules consisted of 96% Eliano MD2 (maltodextrin), 2% tricalcium phosphate and 2% magnesium stearate. Hydroxytyrosol capsules consisted of 63.5% Eliano MD2 (maltodextrin), 2% tricalcium phosphate, 2% magnesium stearate and 34.5% Mediteanox. Mediteanox™ is a natural extract obtained from olive tree (Olea europaea) grown in Spain. This extract is obtained by a patented technology, using extraction procedures based on ultrapure water. The main component of Mediteanox™ is the potent natural antioxidant HT (15 mg/capsule). Toxicology Tests support the safety of Mediteanox™..”

Reviewer: Line 56: VEGF, what does this abbreviation mean?

Authors:  the mean of VEGF abbreviation has been included in the manuscript. “can promote angiogenesis through the regulation of receptors for vascular endothelial growth factor (VEGF) and the activity of matrix metalloproteinases (MMPs)[7]”

Reviewer: The inclusion and exclusion criteria are not presented as they are written in the manuscript. They are usually presented in a paragraph and not separated by periods.

Authors: talking into account this comment, authors have presented exclusion and inclusion criteria in one paragraph. “The following inclusion criteria were considered to participate in the clinical trial: luminal (A or B) tumor phenotype, receiving neoadjuvant therapy based on anthracyclines and taxanes and signed the Informed Consent. The exclusion criteria were: metastatic disease at diagnosis, Her2-positive or triple-negative phenotypes, history of mental illness, detection in the initial interview of a psychological profile that might lead to a low adherence to the supplementation regiment, women who only received hormonal therapy as neoadjuvant treatment and women with known allergies to foods or compounds derived from the olive tree.”

Reviewer: Figure 2 and 3. Plasma level of X ????. Replace with MMP-9 and TIMP-1.

Authors:  According with the reviewer suggestion authors have included plasma level of MMP-9 and plasma level of TIMP-1 in figure 2 and 3 respectively.

Reviewer:  Do the authors have information on the intake of foods with a high HT content (for example, Extra Virgin Olive Oil)? It would be very interesting to include this information.

Authors:  Taking into account this consideration, authors have included the following paragraph: “The daily food intake of the patients decreased due to the chemotherapy treatment. The intake of extra virgin olive oil was recorded in all patients throughout the study (approximately 10-15ml/day in both groups).”.

Reviewer: The authors have data on oxidative stress and plasma antioxidant capacity. It would be very good to complement the obtained results

Authors:  Unfortunately, oxidative stress and plasma antioxidant capacity results from this study have been submitted to other journal and these results have not yet been published.

Reviewer: HT as a cytoprotective agent has been shown to be an efficient regulator of the activity of transcription factors such as Nrf2 and NF-kB in liver and adipose tissue. What can the authors discuss in this regard?

Suggested references:

Illesca et al., Hydroxytyrosol supplementation ameliorates the metabolic disturbances in white adipose tissue from mice fed to high-fat diet through recovery of transcription factors Nrf2, SREBP-1c, PPAR-γ and NF-κB. Biomed Pharmacother. 2019; 109: 2472-2481.

 Valenzuela et al., Molecular adaptations underlying the beneficial effects of hydroxytyrosol in the pathogenic alterations induced by a high-fat diet in mouse liver: PPAR-α and Nrf2 activation, and NF-κB down-regulation. Food Funct. 2017; 8 (4): 1526-1537.

Authors:  authors appreciate the reviewer's suggestion and the discussion has been modified based on Illesca et al and Valenzuela et al publications as follows NFkB is a transcription factor activated in response to bacterial or viral stimuli, growth factor or inflammatory molecules. Activation of NFkB is a common characteristic of many tumors [30]. Moreover, some studies conducted by Illesca et al [31] and Valenzuela et al [32] have concluded that the activation of the transcription factors Nrf2 and PPAR-delta and the down-regulation of NFkB and SREBP-1c appear as important mechanisms mediating the beneficial effect of HT in metabolic disturbances in white adipose tissue. In addition, Liu et al [33] stated that NFkB activation significantly increased the transcription of TNF-alfa, which can elicit NFkB signaling, subsequently causing an up-regulation of TGF-B [30]. TGF-B induces TIMP-1 gene expression in fibroblasts [34]. The above mentioned shows that HT can downregulate NFkB and it could decrease TNF-alfa and TGF-b levels as described for other antioxidant agents, such as silymarin in rat liver injury induced by CCl4 [35] causing a decrease in plasma levels of TIMP-1 in women with breast cancer…”

Reviewer: What is or would be the possible mechanism that would be explaining the observed results?

Authors:  Considering this kind recommendation, a possible mechanism has been included in the manuscript in the discussion issue. “NFkB is a transcription factor activated in response to bacterial or viral stimuli, growth factor or inflammatory molecules. Activation of NFkB is a common characteristic of many tumors [30]. Moreover, some studies conducted by Illesca et al [31] and Valenzuela et al [32] have concluded that the activation of the transcription factors Nrf2 and PPAR-delta and the down-regulation of NFkB and SREBP-1c appear as important mechanisms mediating the beneficial effect of HT in metabolic disturbances in white adipose tissue. In addition, Liu et al [33] stated that NFkB activation significantly increased the transcription of TNF-alfa, which can elicit NFkB signaling, subsequently causing an up-regulation of TGF-B [30]. TGF-B induces TIMP-1 gene expression in fibroblasts [34]. The above mentioned shows that HT can downregulate NFkB and it could decrease TNF-alfa and TGF-b levels as described for other antioxidant agents, such as silymarin in rat liver injury induced by CCl4 [35] causing a decrease in plasma levels of TIMP-1 in women with breast canceret al, 2018) causing a decrease in TIMP-1 plasma level in women with breast cancer...”

Reviewer: What is the projection of the study? Discuss this point.

Authors: the projection of the study would be to obtain a better prognosis and to improve the quality of life in women with breast cancer by supplementing with 15mg / day of hydroxytyrosol during the treatment with chemotherapy due to the decrease in plasma TIMP-1 concentration. Authors believe that this point has been clarified in the revised manuscript.

Minor Comments

Reviewer: Improve the title, because the use of ",". In scientific English it is very little used.

Authors:  According to the reviewer’s suggestion the title has been modified. “Hydroxytyrosol supplementation ameliorates plasma levels of tissue inhibitor of metallopeptidase 1 in women with breast cancer”

Reviewer:  Improve the wording of the study objective

Authors:  Considering this kind recommendation, the objective has been rewritten as follows: “Based on the above, the objective of the present study was to elucidate whether HT improves the antitumor response of women with breast cancer undergoing neoadjuvant chemotherapy (treated with epirubicin and cyclophosphamide followed by taxanes). We also aim to elucidate if this effect could be produced by modulating the plasma levels of MMP-9 and TIMP-1, leading to a decrease in cell proliferation and, therefore, improving the prognosis of the patients.”

Reviewer: Improve the legends of tables and figures

Authors:  Taking into account this consideration, legends of tables and figures have been modified as follows: Table 1: Data are expressed as mean± SEM. BMI: Body Mass Index. p < 0.05 is considered statistically significant; Figure 2. Plasma levels of MMP-9 in women with breast cancer. Values are shown as the mean ± standard error of the mean. Bars not sharing superscript letters are statistically different at p < 0.05 for each experimental group (lowercase for Group A treated with hydroxytyrosol and uppercase for Group B treated with placebo) at each time point (TI, T2 and T3); Figure 3. Plasma levels of the tissue inhibitor of metalloproteinases I (TIMP-1) in women with breast cancer. Values are shown as the mean ± standard error of the mean. Bars not sharing superscript letters are statistically different at p < 0.05 for each experimental group (lowercase for Group A treated with hydroxytyrosol and uppercase for Group B treated with placebo) at each time point (TI, T2 and T3).

Reviewer: Do the authors have information on the tolerance of the product used?

Authors:  Considering this kind recommendation, a new paragraph has been included in materials and methods issue: “All patients took every day the hard capsule containing HT or the placebo capsule during the neoadjuvant chemotherapy until the pre-surgery day (T1, T2 and T3) (Figure 1). No adverse reactions were caused by these capsules and they were well accepted and tolerated by all patients. …”

All authors have seen checked and approved the changes and suggestions exposed in this document.

Reviewer 3 Report

In the present study the others investigated the potential effect of Hydroxytyrosol supplementation in the improves the antitumor response of women with breast cancer undergoing neoadjuvant chemotherapy. They elucidate if this effect could be produced by modulating the plasma levels of MMP-9 and TIMP-1, leading to a decrease in cell proliferation and, therefore, improving the prognosis of the patients. This is a very interesting paper, the other hand this work presents some small corrections according to the following comments.

Comments to Authors

1- Page 3 lin 114: It is necessary to specify the origin (production company) of the Hydroxytyrosol or Mediteanox™ .

2- Modify μl by μL, ml by mL and l by L, in the all of the manuscript.

Author Response

We thank reviewer # 3 for constructive and valuable comments, we have modified the text according to the reviewer suggestions. All changes set out in Article have been highlighted in red ink to make them more visible.

Reviewer: In the present study the others investigated the potential effect of Hydroxytyrosol supplementation in the improves the antitumor response of women with breast cancer undergoing neoadjuvant chemotherapy. They elucidate if this effect could be produced by modulating the plasma levels of MMP-9 and TIMP-1, leading to a decrease in cell proliferation and, therefore, improving the prognosis of the patients. This is a very interesting paper, the other hand this work presents some small corrections according to the following comments.

 1- Page 3 line 114: It is necessary to specify the origin (production company) of the Hydroxytyrosol or Mediteanox™

Authors:  According to the reviewer suggestions, authors have included the origin of Hydroxytyrosol used in the assay as follows: The capsules used in the study were supplied by Probelte Pharma S.A. (Murcia, Spain) following all the quality, stability, conservation and labeling controls for human dietary supplements. Placebo capsules consisted of 96% Eliano MD2 (maltodextrin), 2% tricalcium phosphate and 2% magnesium stearate. Hydroxytyrosol capsules consisted of 63.5% Eliano MD2 (maltodextrin), 2% tricalcium phosphate, 2% magnesium stearate and 34.5% Mediteanox. Mediteanox™ is a natural extract obtained of leaves and olives from olive tree (Olea europaea) of Jaen grown in Spain

Reviewer: Modify μl by μL, ml by mL and l by L, in the all of the manuscript.

Authors:  Taking into account this consideration, authors have changed µl by µL in the manuscript.

All authors have seen checked and approved the changes and suggestions exposed in this document.

Reviewer 4 Report

The average content of hydroxytyroxol in olives is 55-65mg / 100gFW (phenol-explorer database) and the average consumption of olives in the province of Jaén from which patients are recruited is high, an exhaustive count of the consumption of olives and oil should be included of olive in the patients' diet. One serving of 50g olives has 25-30mg of hydroxytyrosol, more than the experimental daily dose. In addition, the consumption of virgin or extra virgin olive oil is common in the province, but the amount you register is very low (10-15ml / day), studies in our research group estimate a consumption in 50ml / day (Ruiz, N. and cols. Antioxidants 2019, 8, 271), and in another study also in Andalusia estimates 36ml / day EVOO for women over 50 years (Amelia de la Torre-Robles and cols. 2014 Estimation of the intake of phenol compounds from virgin olive oil of a population from southern Spain, Food Additives & Contaminants: Part A, 31: 9, 1460-1469, http://dx.doi.org/10.1080/19440049.2014.935961)

Have you registered the menopausal status, alcohol intake, tobacco, and the level of physical activity in the patients, to study whether there is correlation of the MMP9, TIMP1 concentration with these parameters? these could be characteristics of the population which offer heterogeneity of the experimental groups.

In material and methods the description of the method of measurement of MMP9 and TIMP1 is too extensive, only the protocol is followed without modifications of the commercial house?

Author Response

We thank reviewer # 4 for constructive and valuable comments, we have modified the text according to the reviewer suggestions. All changes set out in Article have been highlighted in red ink to make them more visible.

Reviewer: The average content of hydroxytyroxol in olives is 55-65mg / 100gFW (phenol-explorer database) and the average consumption of olives in the province of Jaén from which patients are recruited is high, an exhaustive count of the consumption of olives and oil should be included of olive in the patients' diet. One serving of 50g olives has 25-30mg of hydroxytyrosol, more than the experimental daily dose. In addition, the consumption of virgin or extra virgin olive oil is common in the province, but the amount you register is very low (10-15ml / day), studies in our research group estimate a consumption in 50ml / day (Ruiz, N. and cols. Antioxidants 2019, 8, 271), and in another study also in Andalusia estimates 36ml / day EVOO for women over 50 years (Amelia de la Torre-Robles and cols. 2014 Estimation of the intake of phenol compounds from virgin olive oil of a population from southern Spain, Food Additives & Contaminants: Part A, 31: 9, 1460-1469, http://dx.doi.org/10.1080/19440049.2014.935961)

Authors: The authors agree with the reviewer's comment. However, the intake of olives and olive oil has been recorded during the trial and it is effectively low compared to the healthy population, but we should keep in mind that our patients are being treated with chemotherapy and the total daily food intake decreases considerably compared to a healthy person. This is the reason why the values ​​shown in the manuscript are low compared to the healthy population.

Reviewer: Have you registered the menopausal status, alcohol intake, tobacco, and the level of physical activity in the patients, to study whether there is correlation of the MMP9, TIMP1 concentration with these parameters? these could be characteristics of the population which offer heterogeneity of the experimental groups.

Authors:  The authors agree with the comments of the reviewer but, unfortunately, taking into account the statistical tests, the size of the experimental groups is not sufficient to be able to divide them in each group according to alcohol intake, physical activity, menopause, etc. although these variables have been collected in the study.

Reviewer: In material and methods the description of the method of measurement of MMP9 and TIMP1 is too extensive, only the protocol is followed without modifications of the commercial house?

Authors:  Taking into account this consideration, authors have shorted the description of  MMP9 and TIMP1 measurements in method issue.

All authors have seen checked and approved the changes and suggestions exposed in this document.

Reviewer 5 Report

In brief, clinical trials results are very important and very inclusive data. the presented articles information is too short here. In addition, findings are not very significant. there are lots of confusion remains in the study.  design of the experiments also feels very poor. I am fine with the hypothesis but conducting whole experiments and data collection and data analysis feel poor. its need expertise and careful consideration. it's not just simply any animal observational study.

Specific comments:

Line 16: “vey veried” is not suitable english.

Line 41: if cancers cells DNA has been damaged due to oxidative stress then how tumor initiation progressing?

Line 43: is it possible to mention clearly which enzymes causing maximization and which causing minimizations?

Line 45: can you mention the specific receptor name instead of saying in general? and please do the specific receptor citation

Line 50: tumor progressionàPlease include a specific reference

Line 56: its self oil can not be fractionated until you go column separation. Its better to say " minor amount or quantity"

Line 69: Author says there is no clinical trial but found that: This is a pilot study evaluating the effect of hydroxytyrosol, a component of olive oil, on mammographic density in women at high risk of developing breast cancer. ClinicalTrials.gov Identifier: NCT02068092

Please review again this statement

Line 103: 15 mg/dayà Based on what selected this dose for human study

Line 104: what is the number of patients in each group?

Figure 1:

when started chemotherapy and how many doses they administered should be clearly mentioned. Based on what they started and end the treatment also should be clearly defined. Exactly when the started HT also should be clearly define and how long they continued. when they made surgery and post surgery did they receive any radiation therapy or not also should be define. based on what they select the patients and distribute also should be mentioned. what are the samples were collected and and when collected for the different assays also should be mention in right way.

Line 139:

during the samples collections what was the patient conditions. is the samples always in a particular time or in a random situation. when and how the samples were collected

Figure 2:

how author will ensure it is the effect of HT not the effect of chemotherapy? did author any evidence of single use of HT uses? unify the font and front size

Line 237: yes it is started after chemotherapy with HT but how author will ensure it is the effect for HT not for the chemotherapy.

Figure 3: unify the font and front size

Line 264: then why placebo group also reducing if only HT has the role then placebo group should be maintain high level . assuming that chemotherapy also has role that why over the time of treatment group A and group B become non significant.

Line 280: which means HT has minimal role. the changing of the level is regulating by the chemotherapeutics agents.

Author Response

We thank reviewer # 5 for constructive and valuable comments, we have modified the text according to the reviewer suggestions. All changes set out in Article have been highlighted in red ink to make them more visible.

Reviewer: In brief, clinical trials results are very important and very inclusive data. The presented articles information is too short here. In addition, findings are not very significant. There are lots of confusion remains in the study.  Design of the experiments also feels very poor. I am fine with the hypothesis but conducting whole experiments and data collection and data analysis feel poor. Its need expertise and careful consideration. It’s not just simply any animal observational study.

Specific comments:

Reviewer: Line 16: “vey varied” is not suitable English.

Authors: The authors agree with the reviewer's comment and “very varied” has been changed by different.

Reviewer: Line 41: if cancers cells DNA has been damaged due to oxidative stress then how tumor initiation progressing?.

Authors: the authors want to clarify that this paragraph means that ROS generate oxidative damage in healthy cells contributing to mutagenesis which is essential for the process of tumor initiation. This does not mean that cancer cells are damaged by free radicals.

Reviewer: Line 43: is it possible to mention clearly which enzymes causing maximization and which causing minimizations?

Authors:  According to the reviewer suggestion this paragraph has been changed as follows: This damage caused by free radicals can be minimized by enzymes such as catalase, superoxide dismutase or glutathione peroxidase; or by other non-enzymatic antioxidant mechanisms (vitamins A, C and E, selenium and reduced glutathione (GSH) [4] and maximized by cytochrome P450, xanthine oxidase and NADPH oxidases [5]

Reviewer: Line 45: can you mention the specific receptor name instead of saying in general? and please do the specific receptor citation

Authors:  Taking into account this consideration, authors have mentioned the specific receptor name and they have included a specific receptor citation. As follows:

 In the stage of cancer promotion, ROS can interact with surface or intracellular receptors (tyrosine kinases receptor (RTKs), thus modulating signaling pathways (MAPK- and PI3 Kinase dependent) and physiological mechanisms related to proliferation, apoptosis, angiogenesis and others [6].

Reviewer: Line 50: tumor progression. Please include a specific reference

Authors:  Taking into account this consideration, authors have included a reference in this paragraph.

Reviewer: Line 56: its self oil can not be fractionated until you go column separation. Its better to say " minor amount or quantity"

Authors:  Taking into account this consideration, authors have changed this paragraph as follows:  Hydroxytyrosol (HT) is a polyphenol with a phenethyl alcohol structure. The chemical name of HT is 3,4-dihydroxyphenylethanol. This compound is in minor amount in extra virgin olive oil, in particular, in the water-soluble fraction.

Reviewer: Line 69: Author says there is no clinical trial but found that: This is a pilot study evaluating the effect of hydroxytyrosol, a component of olive oil, on mammographic density in women at high risk of developing breast cancer. ClinicalTrials.gov Identifier: NCT02068092

Authors:  line 69 refers to clinical trials in women with diagnosed breast cancer and not in women with breast cancer risk factor.

Reviewer: Line 103: 15 mg/day  Based on what selected this dose for human study

Authors:  The choice of HT dose for this clinical trial was designed according to

 some manuscripts carried out in healthy people (Marrugat et al. 2004, Covas et al. 2006, Castaner et al. 2012, Oliveras-Lopez et al. 2013, de Bock et al. 2013, Valls et al. 2015, Takeda et al. 2013, Lockyer et al. 2015, Verhoeven et al. 2015) and based on the publication of the European Food Safety Agency (EFSA) that establishes the range of 2 to 15 mg of HT per day, in order to establish a safe dose for patients. [EFSA NDA Panel (EFSA Panel on Dietetic Products, N. a. A. (2011). "Scientific Opinion on the substantiation of health claims related to polyphenols in olive and protection of LDL particles from oxidative damage (ID 1333, 1638, 1639, 1696, 2865), maintenance of normal blood HDL cholesterol concentrations (ID 1639), maintenance of normal blood pressure (ID 3781), “anti-inflammatory properties” (ID 1882), “contributes to the upper respiratory tract health” (ID 3468), “can help to maintain a normal function of gastrointestinal tract” (3779), and “contributes to body defences against external agents” (ID 3467) pursuant to Article 13(1) of Regulation (EC) No 1924/2006." EFSA Journal 9(4): 25].

Reviewer: Line 104: what is the number of patients in each group?

Authors:  the number of patients in each group is included in materials and methods issue as follows:

Patients were randomly assigned to one of the groups. The final number of patients included in the study was 40 (n = 20 per experimental group). The timeline of the study is shown in Figure 1.

Anyway,  authors have included it at the beginning of material and methods as follows: Two experimental groups were established to investigate the effect of dietary supplementation with HT in breast cancer patients undergoing neoadjuvant chemotherapy: patients in Group A (n=20) were supplemented with HT at a single dose of 15 mg / day administered as hard capsules; and patients in Group B (control group n=20) received a placebo hard capsule.

Reviewer: Figure 1: when started chemotherapy and how many doses they administered should be clearly mentioned. Based on what they started and end the treatment also should be clearly defined. Exactly when the started HT also should be clearly define and how long they continued. when they made surgery and post surgery did they receive any radiation therapy or not also should be define. based on what they select the patients and distribute also should be mentioned. what are the samples were collected and and when collected for the different assays also should be mention in right way.

Authors:  Taking into account this consideration, legend of Figure 1 has been modified to clarify the timeline of the clinical study as follows:

Figure 1. Timeline of the clinical study. T1: study start, total time of the period 63 days, 3 cycles of chemotherapy with epirubicin and cyclophosphamide, 21 days each cycle. T2: star of treatment with taxanes, total time of the period 63 days, 3 cycles of chemotherapy, 21 days each cycle. T3: end of chemotherapy treatment and pre-surgery day. HT dose 15mg/d from T1 until T3.

Reviewer: Line 139: during the samples collections what was the patient conditions. is the samples always in a particular time or in a random situation. when and how the samples were collected

Authors:  Taking into account this consideration, the authors have rewritten this part of the material and methods as shown below.

The overall sample size was estimated based on the data obtained in the study published by Vera-Ramírez et al. [24]. Matched blood samples were collected from each patient in T1, T2 and T3 cycles of chemotherapy. Approximately 5ml of blood was taken from each patient, by venous puncture, drawn into an ethylene diamine tetra-acetic acid (EDTA)–containing tube (Vacutainer_EDTA Tubes; BD), and centrifuged at 1000 g for 15 min. The plasma was kept in a separate tube and frozen at - 80_C. The sample was recruited consecutively to its identification and sequentially.

Reviewer: Figure 2: how author will ensure it is the effect of HT not the effect of chemotherapy? did author any evidence of single use of HT uses?

Authors:  I am sorry but authors do not describe that hydroxytyrosol has effect on MMP-9 plasma level: “In this study, no significant differences in plasma levels of MMP-9 were found between the group treated with HT and the group that received the placebo. In contrast, other studies have shown that HT at a dose of 1-10 μmol / L decreased the expression of MMP-9 in cultures of bovine aortic endothelial cells [25] and monocytes, attributing it an antiangiogenic activity [26]. This inconsistency in the results could be explained by the fact that the cited study was performed on a culture of cells from vascular tissue but not on tumor cells”.

Reviewer: Line 237: yes it is started after chemotherapy with HT but how author will ensure it is the effect for HT not for the chemotherapy.

Authors:  authors agree with the reviewer's comments 5 although in our manuscript authors do not show any effect of HT in MMP-9 plasma level.

Reviewer: Line 264: then why placebo group also reducing if only HT has the role then placebo group should be maintain high level. assuming that chemotherapy also has role that why over the time of treatment group A and group B become non significant.

Authors:  The authors agree with the comments of the reviewer 5 although the text remarks that in the two groups after treatment with chemotherapy based on epirubicin-cyclophosphamide, the values of TIMP-1 decreased in both groups but it was only statistically significant for the group treated with HT. However, the statistical non-significance between experimental groups A and B may be due to the fact that the sample size in both groups is not sufficient to detect statistical differences. Another reason would be to stop the treatment with anthracyclines by clinical protocol, perhaps if the treatment is prolonged we could have obtained significant differences. Finally, authors want to emphasize that finding statistical differences inter-subjects is more important than in inter-subjects due to the inter-subject variability.

Reviewer: Line 280: which means HT has minimal role. the changing of the level is regulating by the chemotherapeutics agents.

Authors:  I am sorry but authors have not found the expression minimal role of HT in line 280 or in its paragraph.

All authors have seen checked and approved the changes and suggestions exposed in this document.

Round 2

Reviewer 1 Report

Overall the manuscript has not been improved. In some cases, it is even worse than the previous one, for example  the title is not explicit. What is the meaning of "AMELIORATE?  

Table 1 is still not clear: which is the “Tumor Phenotipe”?

The statistical analysis is still not clear and it seems that no significant physiological relevant effect was observed after treatment with HT.

Author Response

We thank reviewer # 1,2,3,4,5, and Academic Editor for constructive and valuable comments, we have modified the text according to the reviewer’s suggestions. All changes set out in Article have been highlighted in red ink to make them more visible.

Reviewer: Overall the manuscript has not been improved. In some cases, it is even worse than the previous one, for example  the title is not explicit.

Authors:  According to the reviewer suggestions, authors have checked all the manuscript to improve and to clarify it. Authors have changed the manuscript according with all the reviewers (1, 2, 3, 4 and 5) and the Academic Editor suggestions point by point. This means that maybe it is difficult to all agree on the suggestions.

Reviewer: What is the meaning of "AMELIORATE? 

Authors:  considering the reviewer and Academic Editor suggestions the title has been changed as follow:

Hydroxytyrosol supplementation modifies plasma levels of tissue inhibitor of metallopeptidase 1 in women with breast cancer

Reviewer: Table 1 is still not clear: which is the “Tumor Phenotype”?

Authors:  According to the reviewer suggestion our pathologist (C. Ramirez-Tortosa, one of the authors) apologizes because this expression was a confusion when translating it from Spanish to English. So tumor phenotype has been changed by subtypes of breast cancer.

Reviewer: The statistical analysis is still not clear and it seems that no significant physiological relevant effect was observed after treatment with HT.

Authors: Statistical analyzes have been carried out by a statistician. the authors believe that they have been well described in the statistics section on material and methods issue as show below:  the results are expressed as mean and standard error of the mean. Before performing the statistical analysis, the normality and homogeneity of the variances were verified using the Kolmogorov-Smirnoff and Levene tests. When the variance followed a normal distribution and it was homogeneous, the Bonferroni test was used. Whereas, when the variances were not homogeneous, the nonparametric Kruskall-Wallis, U-Mann-Whitney and Tamhane tests were used. An analysis of the variance (ANOVA) was performed to find out if there were differences among time points (T1, T2 and T3) for each experimental group (A and B). The significance was established at p <0.05. The Student’s t test was performed to find differences between Group A and Group B for the same time point. All the statistical analyses were performed using the SPSS software version 22.0 (IBM Corp., Armonk, NY, USA). The results are shown in bar graphs. Bars not sharing superscript letters are statistically different (p< 0.05). Moreover, authors according with other reviewer suggestions have rewritten some paragraph in results issue to clarify them.

All authors have seen checked and approved the changes and suggestions exposed in this document.

Reviewer 2 Report

The authors made all the suggested changes. I accept the manuscript in its current version.

Author Response

We thank reviewers # 1,2,3,4 and 5 and de Academic Editor for constructive and valuable comments, we have modified the text according to the reviewer suggestions. All changes set out in Article have been highlighted in red ink to make them more visible.

Reviewer: English language and style are fine/minor spell check required

Authors:  According to the reviewer suggestions, authors have checked all the manuscript to improve the English grammar and to clarify it.

All authors have seen checked and approved the changes and suggestions exposed in this document.

Reviewer 5 Report

Author addressed issue during the review. The overall article has a good impact on the readers.